# Preparation and Characterization of Fe-Mn Binary Oxide/Mulberry Stem Biochar Composite Adsorbent and Adsorption of Cr(VI) from Aqueous Solution

**DOI:** 10.3390/ijerph17030676

**Published:** 2020-01-21

**Authors:** Meina Liang, Shuiping Xu, Yinian Zhu, Xu Chen, Zhenliang Deng, Liling Yan, Huijun He

**Affiliations:** 1College of Environmental Science and Engineering, Guilin University of Technology, Guilin 541004, China; liangmeinaa@163.com (M.L.); shuipingxuu@163.com (S.X.); benrenchenxu@163.com (X.C.); dengzhenliang088@163.com (Z.D.); yanlilingg@163.com (L.Y.); hehj@glut.edu.cn (H.H.); 2Guangxi Key Laboratory of Environmental Pollution Control Theory and Technology, Guilin 541004, China

**Keywords:** mulberry stem, bio-charcoal, adsorption, Fe-Mn binary oxide, chromium

## Abstract

This study details the preparation of Fe-Mn binary oxide/mulberry stem biochar composite adsorbent (FM-MBC) from mulberry stems via the multiple activation by potassium permanganate, ferrous chloride, triethylenetetramine, and epichlorohydrin. The characteristics of FM-MBC had been characterized by SEM-EDS, BET, FT-IR, XRD, and XPS, and static adsorption batch experiments such as pH, adsorption time, were carried out to study the mechanism of Cr(VI) adsorption on FM-MBC and the impact factors. The results indicated that in contrast with the mulberry stem biochar (MBC), the FM-MBC has more porous on surface with a BET surface area of 74.73 m^2^/g, and the surface loaded with α-Fe_2_O_3_ and amorphization of MnO_2_ particles. Besides, carboxylic acid, hydroxyl, and carbonyls functional groups were also formed on the FM-MBC surface. At the optimal pH 2.0, the maximum adsorption capacity for Cr(VI) was calculated from the Langmuir model of 28.31, 31.02, and 37.14 mg/g at 25, 35, and 45 °C, respectively. The aromatic groups, carboxyls, and the hydroxyl groups were the mainly functional groups in the adsorption of Cr(VI). The mechanism of the adsorption process of FM-MBC for Cr(VI) mainly involves electrostatic interaction, surface adsorption of Cr(VI) on FM-MBC, and ion exchange.

## 1. Introduction

With the rapid development of industrialization, chromium-containing compounds has been extensively used in industrial production, such as tanning, electroplating, dyeing, and coloring pigments [1,2]. China is one of the primary chromium producing countries. There are about 25 firms manufacturing chromium salts, and the chromium salts of these firms could reach 329,000 tons every year [3]. Chromium principally exists in Cr(III) and Cr(VI) oxidation states [4,5]. The toxicity of Cr(VI) is five hundred times that of Cr(III). Cr(VI) may cause teratogenesis, mutation, or carcinogenesis for living creatures [6] and has been chosen as a priority pollutant by the USEAP [7]. The maximum concentrations of Cr(VI) set by the Environmental Protection Agency in industrial wastewater and drinking water were 200 and 50 ug/L, respectively [8].

Currently, the main techniques for removing chromium include ion exchange, membrane separation, electrolytic removal, chemical reduction, and adsorption [8]. Among these methods, as one of the most promising techniques, adsorption technology has been employed for a several decade years, and the effectiveness has been demonstrated [9]. Biochar has less porosity and more surface oxygen-containing functional groups than activated carbon [10]. These characteristics show that biochar has a strong ion exchange capacity and has been used to adsorb organic contaminants, heavy metals, and other pollutants and for environmental remediation, etc. [11,12,13]. Guangxi province is located in the southeast China. The annual output of mulberry stem in Guangxi is about 1 million tons, which are frequently burned due to share utilization technology of mulberry stem [14]. Once mulberry stem was burned in the fields, which not only caused air pollution but also caused resource waste. Mulberry is rich in cellulose and hemicellulose, it can be used as an important feedstock for biochar production because mulberry stem is an adequate, renewable, and low-cost biomass.

Because of the high adsorption affinity and special surface activity, manganese and iron oxide powders have been extensively studied and have been widely used to remove heavy metals [15]. MnOx are also significant scavengers to various ions in contaminated water [16]. Iron oxide has been used in most studies for the modification of biochar or other materials, since it has a natural affinity to chromium [17]. Besides, ferrimanganic binary oxide adsorbents have been widely targeted for removing heavy metals from water due to their valid performance, environmentally friendly properties, and low cost [18]. Wang et al. [19] discovered that FMBC (Biochar impregnated with iron and manganese oxides) indicated improved sorption of Cr(VI) compared with biochar. Iron manganese bimetal oxide nanospheres were synthesized via a facile and environmentally friendly template-free approach, and the maximum adsorption capacities of Cr(VI) was 105.96 mg/g [20]. Du et al. [21] prepared a granular Fe-Mn binary oxide and used it to remove Cr(VI) from aqueous solution, and the maximum adsorption abilities for Cr(VI) was higher than other granular adsorbents for Cr(VI) adsorption. Moreover, Fe-Mn binary oxide adsorbents can increase the pH_zpc_ values of biochar and strongly dominate the charge properties. These are beneficial for the adsorption of chromium in the anionic state [22]. Given all the analysis above, the Ferro-manganese binary oxide with biochar can combine the advantages of manganese and iron oxides, showing strong adsorption capacity for heavy metal. 

The purpose of this study is to assess the adsorption capacity of FM-MBC and MBC to Cr(VI), addressing the following specific objectives: (i) preparation and characterization of FM-MBC and MBC and (ii) investigation of the adsorption properties and adsorption mechanism of Cr(VI) by FM-MBC.

## 2. Experimental Part

### 2.1. Chemical Reagents and Solutions

Reagents purchased from Shanghai Guoyao Group Chemical Reagent Co. Ltd., China, including FeCl_2_•4H_2_O, MgCl_2_•6H_2_O, KMnO_4_, K_2_Cr_2_O_7_, HNO_3_, NaOH, Polyethylene glycol (PEG), triethylenetetramine, etc., were analytical reagent grade. The standard solution of Cr(VI) (1000 mg/L) was purchased from National Nonferrous Metals and Electronic Materials Analysis and Testing Center, China, stored at 4 °C. Conductivity of the experimental water greater than 18.2 MΩ cm (Millipore-Q Direct 8) was used throughout. 

### 2.2. Preparation of FM-MBC

The preparation of Mulberry stem biochar (MBC) and FM-MBC was based on the method reported by Liang et al. (China Patent No. 201811047535.9). The peeled mulberry was smashed about a particle size of 1 mm and then was dried in oven at 80 °C for 24 h. The dry mulberry was carbonized/activated in a muffle furnace at 500 °C for 3 h to get the MBC, and then, it was sieved to a particle size of about 0.15 mm.

A total of 3 g of MBC was added into a 500 mL conical flask; then, 50 mL 0.2 mol/L FeCl_2_ aqueous solution and 0.05 M MgCl_2_ aqueous solution were added into the conical flask rapidly under magnetic stirring for 10 min, and the conical flask was covered with a glass surface dish to impregnate for 3 h. Then 1 mL PEG, 100 mL 0.05 M KMnO_4_ solution, 20 mL 2.5 M NaOH solution and 10 mL epichlorohydrin were added to the mixture suspended solution under magnetic stirring in turn. Under magnetic stirring, the mixture suspended solution continued to react for 3 h at 60 °C. Then 5 mL triethylenetetramine was added to the mixture suspended solution and continued to react for 6 h at 80 °C. After reaction, the suspended solid was cooled, then was filtered, and dried in oven at 65 °C to get the FM-MBC. Finally the FM-MBC was sifted by a 100 mesh sieve.

### 2.3. Characterization 

The structural feature of MBC and the FM-MBC was observed using scanning electron microscopy (SEM, JSM-7900F, Joint-stock Company, Tokyo, Japan). Powder X-ray diffractometer equipped was used for the XRD analysis (X’Pert PROX, PANalytical, B.V., EA Almelo, Netherlands). Infrared spectra were measured using a PE CAT500A FTIR spectrometer (PerkinElmer, Liantrisant, UK). X-ray photoelectron spectroscopy (XPS) experiments of FM-MBC were performed by ESCALAB•250Xi (Thermo Electron Corporation, Massachusetts, USA). The zeta potential of materials was undertaken using a Nano-ZS90 apparatus (Malvern Panalytical, Worcestershire, UK) [23]. The specific surface area of MBC and FM-MBC were determined according to the BET equation by nitrogen gas adsorption using a JW-BK200C apparatus (Beijing Jingwei Gaobo science and Technology Co., Ltd. No.12 Kechuang 13th Street, Beijing Economic and Technological Development Zone, Beijing, China). The content of C, H, N, and S of samples was determined by Perkin Elmer Series ⅡCHNS/O Analyzer 2400 (PerkinElmer, Shelton, USA).

### 2.4. Batch Experiment

The adsorption procedure was as follows, 0.100 g FM-MBC was added in 100 mL polyethylene plastic centrifuge tube; then, 50.0 mL Cr(VI) containing solution was added. The pH value of the solutions was adjusted by 0.1 M sodium hydroxide solution and/or nitric acid solution. The tubes were shaken at desired temperature and time at 200 rpm in a water bath oscillator. Then the mixture was separated by centrifugation, and supernatant was carefully filtered; then, the concentration of Cr(VI) was measured using diphenylcarbazide spectrophotometry.

The Effect of pH was studied from 2.0 to 11.0 (pH 2.0, 3.0, 4.0, 5.0, 6.0, 7.0, 8.0, 9.0, 10.0, 11.0) and the Cr(VI) containing solutions (10 or 20 mg/L) mixed with FM-MBC and shaken for 24 h at 25 °C. To obtain the adsorption isotherms, solutions of various Cr(VI) concentrations (2, 5, 10, 20, 30, 40, 50, 60, 80, 100, 120, 150 mg/L) were shaken with the FM-MBC for 24 h. The effect of the contact time of Cr(VI) adsorption onto FM-MBC was carried out in 50 mL of 10 and 20 mg/L Cr(VI) solution, and the concentration of Cr(VI) was analyzed at different time (0.25, 0.5, 1, 2, 3, 4, 5, 6, 8, 10, 12, 15, 18, 21, 24, 27, 30, 36, 48 h).

## 3. Results and Discussion

### 3.1. Characterization of FM-MBC

#### 3.1.1. Elemental Analysis and Specific Surface Area

Elemental analysis results (Appendix A) indicated that accounting carbon elements were 77.4% and 57.29% in the MBC and FM-MBC, respectively. Oxygen element accounting in MBC increased from 8.94% to 28.31% in FM-MBC; maybe some surface oxygen functional groups were formed on the FM-MBC surface [18]. The specific surface area of FM-MBC was 74.73 m^2^/g by the nitrogen adsorption–desorption method (BET-N_2_). The pore volume of FM-MBC was 0.094 cm^3^/g.

#### 3.1.2. SEM Analysis

The SEM of MBC and FM-MBC was shown in Figure 1; it was found that the MBC had smooth round shape and pore structure. The pore diameter of FM-MBC was between 0.5 and 5.0 um, which was smaller than the pore diameter of MBC. The FM-MBC possessed more porous on the surface in contrast with the MBC and was loaded with iron and/or manganese oxide particles. Moreover, no collapse or failure on the wall surface of pore of FM-MBC was found. However, the surface of the FM-MBC became rough, and micropores with a pore diameter of less than 1 um were increased. The micropore was beneficial to the internal diffusion of adsorbents for heavy metal ions [24]. Numerous spherical dots were added to the inner wall of the pore and attached to the pore surface. These spherical dots may be metal oxides, which can provide more adsorption binding sites.

#### 3.1.3. XRD Analysis 

The XRD patterns of the MBC and FM-MBC were shown in Figure 2. MBC exhibited one broad and gentle amorphous diffraction peak around at 2θ = 24.2°; this was mainly the diffraction peak of crystalline carbon fibers. A strong diffraction peak was recorded at 2θ = 29.5° and exhibited some weak diffraction peaks at 36.1°, 39.4°, 43.3°, 47.6°, and 49.5° of 2θ value, which may be that the MBC was not cleaned and contained Ca, K, and Mg (Appendix A). When MBC was loaded by Fe-Mn binary oxide, some weak diffraction peaks at 36.1°, 39.4°, 43.3°, 47.6°, and 49.5° of 2θ value disappeared, but a new weak diffraction peak at 35.5° was found. The new peak corresponded to α-Fe_2_O_3_ standard peak (110) (PDF NO: 00-033-0664). Compared with MBC, no intensity crystalline typical peaks of MnO_2_ were found of FM-MBC, which represented the formation of the amorphous phase of the FM-MBC [25,26].

### 3.2. Effect of Solution pH

The pH value of the solution is a key factor affecting the removal of Cr(VI) [27], and the surface charge and dissociation of functional groups of biochar [28]. The amount adsorption of Cr(VI) gradually decreased as the pH value increased from 2.0 to 11.0. The optimal pH for Cr(VI) adsorption was 2.0, indicating that an acidic condition is favorable for the removal of Cr(VI). As shown in Appendix A, the pH_zpc_ (point of zero charge) of FM-MBC was 7.4, so when the pH < pH_zpc_, the surface of FM-MBC was positively charged, which was favorable for Chromium-containing anions were adsorbed on FM-MBC. Chromium-containing anions were mainly presented as HCrO_4−_ and Cr_2_O_7_^2−^ in solution at pH range from 2.0 to 6.4 and as CrO_4_^2−^ at pH > 6.4 [29]. Cr(VI) anions were adsorbed onto the positively charged FM-MBC surface by electrostatic interaction. When pH > pH_zpc_, the surface of FM-MBC is negatively charged and caused electrostatic repulsion between the surface of FM-MBC and Chromium-containing anions, which has an adverse effect on Cr(VI) adsorption. At the same time, the OH- occupying a part of the surface active adsorption site of the composite adsorbent to some degree, resulting in competitive adsorption of Cr(VI). This process reduced the removal effect of Cr(VI) by the adsorbent [30].

### 3.3. Effect of Adsorption Time

The effect of time on the adsorption of Cr(VI) on FM-MBC was shown in the Figure 3b. In terms of q_e_-t relationship, the adsorption amount of Cr(VI) increased rapidly at the beginning of the adsorption experiment, indicating favorable interactions between FM-MBC and Cr(VI). The results in Figure 3b also showed that when the initial Cr(VI) concentration is 20 and 50 mg/L, the Cr(VI) adsorption amounts reached equilibrium at 2 and 3 h, respectively, at 25 °C and pH 2.0. It may be due to the fact that when Cr(VI) adsorption capacity was low in the initial reaction stage, the surface of the adsorbent contained numerous adsorption sites, and the adsorption reaction occurred rapidly [31]. The existence of lots of vacancies in the initial stage may be the reason for the rapid initial absorbcion [32].

### 3.4. Adsorption Kinetics

In this study, the Pseudo-first-order kinetic equation, Pseudo-second-order kinetic equation, Banghamkinetic equation, and Elovich kinetic equation were adopted to describe the adsorption kinetic data:

Pseudo-first-order Equation:(1)$$\ln\left(q_{e}-q_{t}\right)={\mathrm{lnq}}_{e}-K_{1}t$$

Pseudo-second-order Equation:(2)$$\frac{t}{q_{t}}=\frac{1}{k_{2}q_{e}^{2}}+\frac{t}{q_{e}}$$

Bangham kinetic Equation:(3)$$\mathrm{lg}q_{t}=\mathrm{lg}k_{3}+\frac{1}{m}\mathrm{lg}t$$

Elovich kinetic Equation:(4)$$q_{t}=a+k_{4}\ln t$$
where q_e_ and q_t_ are the adsorption capacities of the adsorbent at equilibrium and at time t (min), respectively; k_1_ (min^−1^), k_2_ (g/mg·min), k_3_ and k_4_ are the Pseudo first-order kinetic constant, the Pseudo-second-order kinetic constant, the Bangham kinetic constant, the Elovich kinetic constant, respectively.

The fitting plots are shown in Figure 4 and the kinetic parameters acquired from fitting results are summarized in Table 1. The higher correlation coefficient (R^2^) for the Pseudo-second-order equation implied that the adsorption processes of FM-MBC were controlled by chemisorption on their surface [33]. In addition, when the initial concentrations were 20 mg/L and 50 mg/L, the theoretical equilibrium adsorption amounts of Cr(VI) were 9.68 mg/g and 24.27 mg/g at 25 °C, respectively, which was very close to the actual measured equilibrium adsorption amounts (9.72 mg/g and 24.39 mg/g), indicating that these kinetic data conformed to the pseudo-second-order equation very well.

Kinetic Parameters

In the initial Cr(VI), concentration was 50 mg/L, and the influence of temperature on the FM-MBC adsorption Cr(VI) was investigated at 298 and 318 K. The Arrhenius equation was used to calculate activation energy (E_a_) of adsorption as follows [34]:(5)$${\mathrm{lnK}}_{d}=\mathrm{lnA}-\frac{E_{a}}{RT}$$
(6)$$E_{a}=2.303R\left(\frac{T_{1}T_{2}}{T_{1}-T_{2}}\right)l_{g}\frac{K_{1}}{K_{2}}$$
where A (g/mol·min), E_a_ (kJ/mol), R(8.3145 J/mol·K), T (K), K_d_ (g/mg·min), are the independent temperature factor, the activation energy of adsorption, the gas law constant, the solution absolute temperature and reaction rate constant of the pseudo-second-order equation, respectively. The activation energy is indicator for adsorption type, which for a physical adsorption E_a_ value lower than 40 kJ/mol and a chemisorption adsorption is higher than 40 kJ/mol. E_a_ was found to be 4.756 kJ/mol (E_a_ was calculated by Equation (6) with k_2_ in Table 1 at 298 and 318 K). Base on the correlation coefficient (R^2^) for the Pseudo-second-order and E_a_ value, it indicated that the FM-MBC adsorption Cr(VI) was a physical-chemical adsorption process [35].

### 3.5. Adsorption Isotherm

The adsorption isotherm of the MBC and FM-MBC for Cr(VI) (Figure 5), which indicated that the adsorption capacity of Cr(VI) rapidly increased, with increase in temperature at the equilibrium Cr(VI) concentration less than 10 mg/L. When the equilibrium concentration of Cr(VI) was greater than 10 mg/L, the adsorption capacity of Cr(VI) slowly increased. It may be due to the adsorbent not attaining adsorption saturation when the equilibrium concentration was lower and the adsorption reaction occurring on the surface of the adsorbent. The amount of Cr(VI) adsorbed increased from 0.20 to 28.31, 9.78 to 31.02, and 9.83 to 37.54 mg/g when the Ce_(Cr(VI))_ was increased from 0.01 to 43.52 mg/L, at 25, 35, and 45 °C, respectively. As the temperature increased from 25 to 45 °C, the amount of Cr(VI) adsorbed increased from 28.31 to 37.54 mg/g, indicating that increasing temperature was beneficial to Cr(VI) removal, and FM-MBC exhibited a Cr(VI) adsorption capacity higher about one point five times than that of MBC.

In this study, the Langmuir models and Freundlich models were applied to simulate the the experimental equilibrium data of the adsorption of Cr(VI) on the FM-MBC at of 25, 35, and 45 °C.

Langmuir Isotherm

Langmuir model can be represented by the equation as follows:(7)$$\frac{C_{e}}{q_{e}}=\frac{1}{\left(Q_{\max}\cdot K_{L}\right)}+\frac{C_{e}}{Q_{\max}}$$
where Ce means the Cr(VI) concentration of solution (mg/L) at adsorption equilibrium. The Q_max_ represents the adsorption amount (mg/g), and K_L_ is the Langmuir constant (L/mg). The linear graph of plot of C_e_/q_e_ versus C_e_ is shown in Figure 6. The values of Q_max_ and K_L_ are calculated based on the slope and intercept of the linear graph (Table 2).

The results of the fitting process in Figure 6 and Table 2 showed that Langmuir isotherm model is suitable for fitting the adsorption of chromium (VI) on FM-MBC. At 25, 35, and 45 °C, the correlation coefficients (R^2^) of Langmuir isotherm model are all 0.999. The results showed that homogeneous adsorption existed on the FM-MBC surface. The maximum adsorption capacity can be increased from 28.49 mg/g at 25 °C to 37.62 mg/g at 45 °C by increasing the temperature. The results showed that Cr(VI) adsorption on FM-MBC is endothermic. The adsorption capacity of FM-MBC for Cr(VI) at 25 °C is 28.49 mg/g, which is higher than 7.08 mg/g of municipal sludge biochar and 8.05 mg/g of yeast biochar [36,37], slightly higher than 21.45 mg/g of eucalyptus corm biochar [38].

Freundlich Isotherm

Freundlich model can be represented by the equation as follows:(8)$$\ln q_{e}=\ln K_{F}+(\frac{1}{n})\ln C_{e}$$

In the formula, q_e_ is the adsorption capacity of Cr(VI) in equilibrium (mg/g), Ce is the equilibrium Cr(VI) concentration (mg/L), K_F_ (mg/g)/(mg/L)^n^ is Freundlich constant, and 1/n is adsorption strength.

The K_F_ and 1/n values can be calculated from the plot of lnq_e_ versus lnC_e_ (Figure 7, Table 2). At 25, 35, and 45 °C, the correlation coefficients (R^2^) of the Freundlich isotherm model were 0.875, 0.866, and 0.871, respectively, and the Freundlich isotherm could not well fit Cr(VI) adsorption on FM-MBC. 1/n can be used for measuring the deviation from linear of the adsorption [39,40]. In this study, 1/n values were 0.357 (25 °C), 0.350 (35 °C) and 0.341 (45 °C), which indicated that Cr(VI) adsorption onto FM-MBC process belong to be chemical adsorption [39].

Thermodynamic Parameters

Thermodynamic parameters can be calculated using Equations (9) and (10) [41].
ΔG° = −RTlnK_L_(9)
ΔG° = −RTlnK_L_ = ΔH° − TΔS°(10)
where ΔG° is changes in standard free energy; ΔH° is changes in standard enthalpy; ΔS° is changes in standard entropy; R, K_L_, and T are gas constant, the Langmuir constant and the temperature in Kelvin (K), respectively. 

ΔG° can be calculated by Equation (9). ΔH° and ΔS° can be calculated from the plot of ΔG° versus T. As shown in Table 2, the adsorption capacity of Cr(VI) increased with the temperature range from 25 to 45 °C, which indicated endothermic reaction occurred. ΔG° calculated by Equation (9) were −0.252, −0.499, and −0.528 kJ/mol at 25, 35, and 45 °C, respectively, which indicated that the adsorption for Cr(VI) on FM-MBC was spontaneous nature. At the same time, the positive ΔH° values (3.946 kJ/mol) suggested endothermic reaction occurred. Besides, the positive ΔS° values (0.1045 kJ/(mol K)) indicated that the randomness increased at the interface of the solid/solution during the adsorption of Cr (VI) on the FM-MBC [42].

### 3.6. Mechanism of Cr(VI) Adsorption by FM-MBC

#### 3.6.1. SEM-EDS Analysis

The EDS analysis indicated that Mn, Fe, and O were presented in the FM-MBC powder before and after Cr(VI) adsorption, and Cr was presented in the FM-MBC powder after Cr(VI) adsorption (Figure 7). The mass percentages of elements were undertaken at three points (Table 3). According to the results of EDS spectrum, the means of mass percentages (%) of C, O, Mn, and Fe of the FM-MBC before and after adsorption Cr(VI) were 24.33, 28.10, 7.62, 38.01 and 13.89, 4.21, 6.42, 49.32, respectively, and the means of mass percentages (%) of Cr of the FM-MBC after adsorption Cr(VI) was 22.32%. This pattern confirmed that hexavalent chromium was adsorbed on FM-MBC surface.

#### 3.6.2. FTIR Analysis

The FTIR spectra of MBC, FM-MBC and FM-MBC after chromium adsorption were shown in Figure 8. Absorption band at the wave number (ν) values 3419 cm^−1^ was present in three spectra, which indicated the stretch and bending vibration modes of -OH [43]. The band at ν = 2894 cm^−1^ corresponded to the -CH_2_ deformation vibration [44], and the band at ν = 1622 cm^−1^ corresponded to the vibration of aromatic groups (e.g., C = C) and carboxyl (C = O) [45]. Compared to MBC, some new bands at ν = 1428, 1162, 1049, 876, 697, and 566 cm^−1^ were observed in FTIR spectra of FM-MBC. The bands at ν = 1428, 1162, and 1049 cm^−1^ corresponded to the bending vibration of the Fe-OH and Mn-OH [40,46]. Meanwhile, the band at ν = 876 cm^−1^ could be due to Fe-OH vibrations of α-FeOOH. Besides, absorption band at the wave number (ν) values 697 and 566 cm^−1^ could be ascribed to α-Fe_2_O_3_ [40]. However, the bands at ν = 450, 520, and 720 cm^−1^ due to the stretching vibrations of Mn-O were not found in the FTIR spectra of the FM-MBC. It could be the amorphization of MnO_2_ in the FM-MBC, which corresponded to XRD characterization of FM-MBC [25]. The adsorption bands at ν = 1622 and 1162 cm^−1^ became weakened in the FM-MBC after Cr(VI) adsorption than FM-MBC, and it could be seen that the band at ν = 1051 cm^−1^ of the FM-MBC shift from 1049 cm^−1^ to ν = 1051 cm^−1^ after Cr(VI) adsorption, which indicated that aromatic groups, carboxyls, and the hydroxyl groups associated with Fe and Mn involved in the adsorption process [40,46,47].

#### 3.6.3. XPS Analysis

XPS spectrum of MBC and FM-MBC before and after adsorption Cr(VI) were shown in Figure 9. The photoelectron signals of MBC were C1s (284.05 eV) and O1s (531.52 eV). The photoelectron signals of FM-MBC and FM-MBC after Cr(VI) adsorption were C1s (284.40 eV), O1s (530.80 eV), Mn2p (640.90 eV), and Fe2p (710.50 eV). Because the strength of XPS is related to the content of elements, the content of Mn in FM-MBC before and after adsorption in Table 3 is only 6.42–7.62 wt. %, so the peak of Mn in XPS is not obvious. The peak center at 576.7 eV (spectrum FM-MBC after Cr(VI) adsorption) is due to the Cr2p, indicating that Cr(VI) was absorbed onto FM-MBC. This result was consistent with the results of FT-IR spectra of FM-MBC and the results of Yu et al. [48].

High-resolution XPS scans of the Fe2p orbital and Mn2p orbital on the FM-MBC surface before and after adsorption in the range of 700–740 eV and 635–660 eV were undertaken, respectively (Figure 10 and Appendix A). The main Fe2p peak was shifted from 710.55 to 710.76 eV after Cr(VI) adsorption, which suggested iron oxide was involved in Cr(VI) adsorption onto FM-MBC. The binding energy of Mn2p was decreased from 640.48 to 641.03 eV after Cr(VI) adsorption. These values were typical of the valence state of Mn^4+^, which existed in the form of MnO_2_ [49]. These results indicated that both Fe2p and Mn2p were involved in the adsorption reaction.

On the surface of FM-MBC, there are a large number of functional groups on the surface of C-OH, Fe-OH, and Mn-OH (represented by S-OH); and –COOH (represented by S-COOH), when the pH = 2.0 < pH_zpc_, the surface of FM-MBC was positively charged. The combination reaction between FM-MBC and Cr (VI) ions in solution with different forms may be assumed as follows (Equations (11)–(17)).
S-OH + H^+^ → S-OH_2_^+^(11)
S-OH_2_^+^ + HCrO_4−_ → S-HCrO_4_ + H_2_O(12)
S-OH_2_^+^ + Cr_2_O_7_^2−^ → S-Cr_2_O_7_^2−^ + H_2_O(13)
S-OH + HCrO_4−_ → S-HCrO_4_ + OH^-^(14)
S-OH + CrO_4_^2−^ → S-CrO_4−_ + OH^-^(15)
S-COOH + HCrO_4−_ + H_2_O → S-CO_3_H_4_O_3_Cr+ S-OH^−^(16)
S-COOH + CrO_4_^2−^ → S-CO_5_Cr^−^ + S-OH^−^(17)

Moreover, it was supposed that Cr(VI) could adsorb onto the FM-MBC including electrostatic interaction, surface adsorption of Cr(VI) on FM-MBC, and the ion exchange between the surface functional groups of the FM-MBC with Cr(VI). The possible adsorption mechanism is illustrated in Figure 11.

## 4. Conclusions

There is nowadays a growing incentive for fit-for-purpose treatment methods to be developed for metal-containing wastewaters using low-cost materials. This study highlighted that Mulberry stem, an abundant biomass, can serve as a potential cheap source for the preparation of sorbents for hexavalent chromium removal. Loaded with α-Fe_2_O_3_ and amorphization of MnO_2_ particles, FM-MBC had a microporous structure with a BET surface area of 74.73 m^2^/g, and the point of zero charge values (pH_zpc_) of FM-MBC was 7.4. Major carboxylic acid, hydroxyl, and carbonyls were present on the surface of FM-MBC. The optimum Cr(VI) adsorption occurred at pH 2.0. The pH of the optimal adsorption was acidity, which was the only disadvantage of the FM-MBC. At 25, 35, and 45 °C, the correlation coefficients of the Langmuir isotherm adsorption model were 0.999, 0.998, and 0.999, respectively, and the maximum adsorption capacities of Cr(VI) were 28.31, 31.02, and 37.14 mg/g, respectively. These results indicated that monolayer adsorption by the FM-MBC occurred. The process of FM-MBC adsorption Cr(VI) was a spontaneous and endothermic physical-chemical adsorption. The aromatic groups, carboxyls, and the hydroxyl groups associated with Fe and Mn were included in the adsorption process. The main mechanism of the Cr(VI) adsorption on FM-MBC mainly involved electrostatic interaction, surface adsorption of Cr(VI) on FM-MBC, and ion exchange. 

## Figures and Tables

**Figure 1 ijerph-17-00676-f001:**
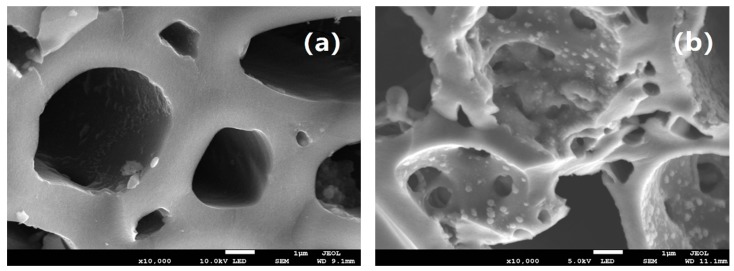
SEM image of mulberry stem biochar (MBC) and Fe-Mn binary oxide/mulberry stem biochar composite adsorbent (FM-MBC). (**a**) MBC, (**b**) FM-MBC.

**Figure 2 ijerph-17-00676-f002:**
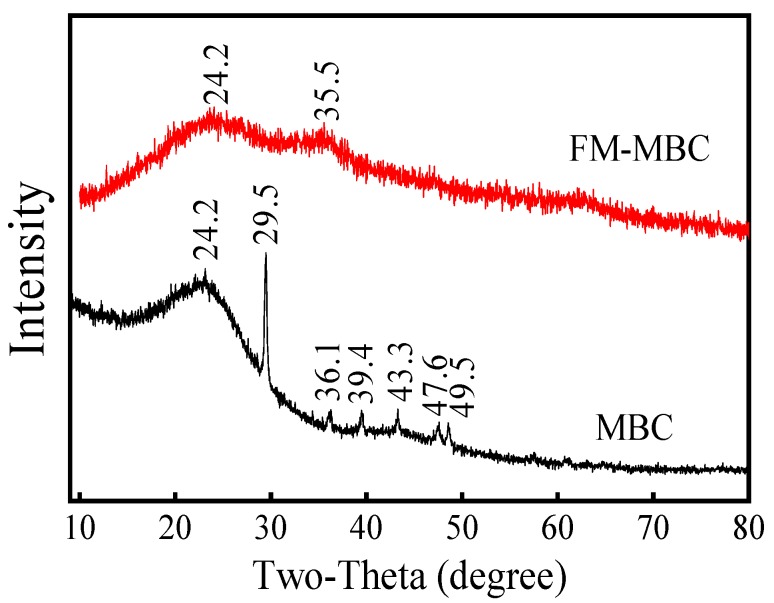
XRD spectrum of MBC and FM-MBC.

**Figure 3 ijerph-17-00676-f003:**
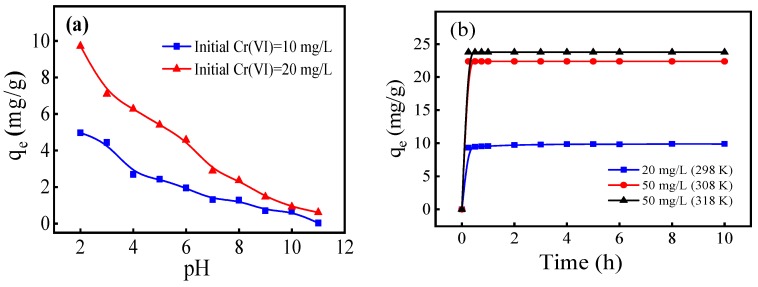
The effect of pH and contact time on Cr(VI) adsorption onto FM-MBC. (**a**) pH, (**b**) contact time.

**Figure 4 ijerph-17-00676-f004:**
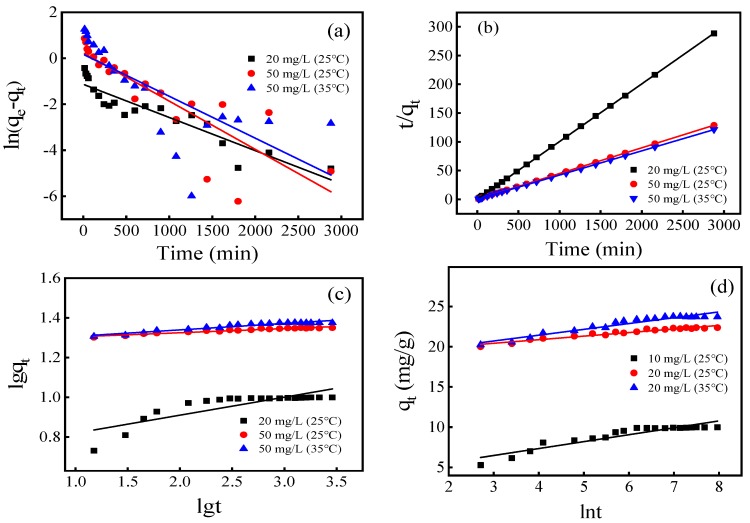
Adsorption kinetics of Cr(VI) adsorption on FM-MBC. (**a**) Pseudo-first-order, (**b**) Pseudo-second-order, (**c**) Banghamkinetic, (**d**) Elovich kinetic.

**Figure 5 ijerph-17-00676-f005:**
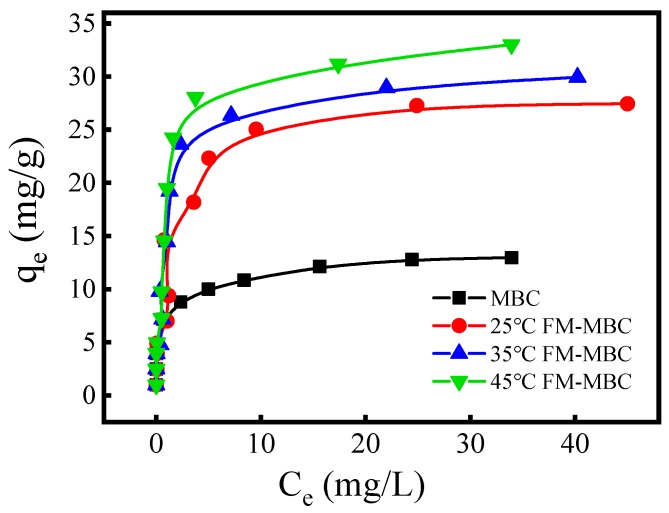
Adsorption isotherms of Cr(VI) adsorption on FM-MBC and MBC.

**Figure 6 ijerph-17-00676-f006:**
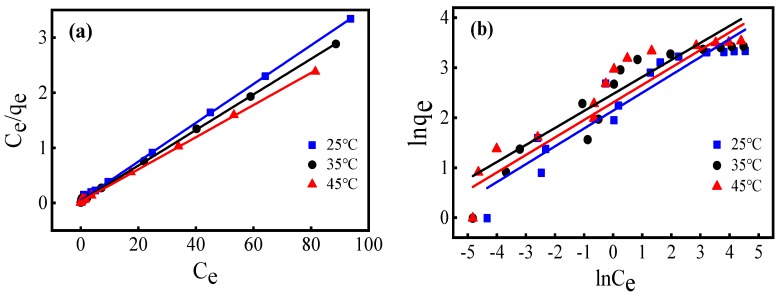
Isotherm plots for Cr(VI) adsorption onto FM-MBC. (**a**) Langmuir equation, (**b**) Freundlich equation.

**Figure 7 ijerph-17-00676-f007:**
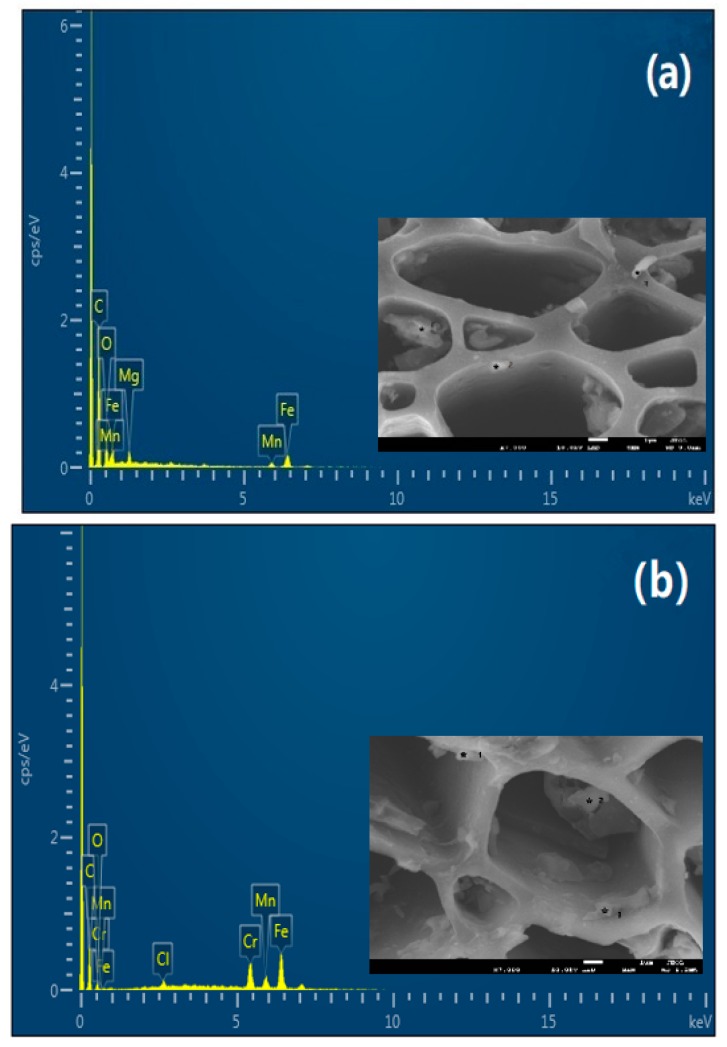
SEM-EDS spectrum of the FM-MBC. (**a**) FM-MBC, (**b**) FM-MBC after Cr(VI) adsorption.

**Figure 8 ijerph-17-00676-f008:**
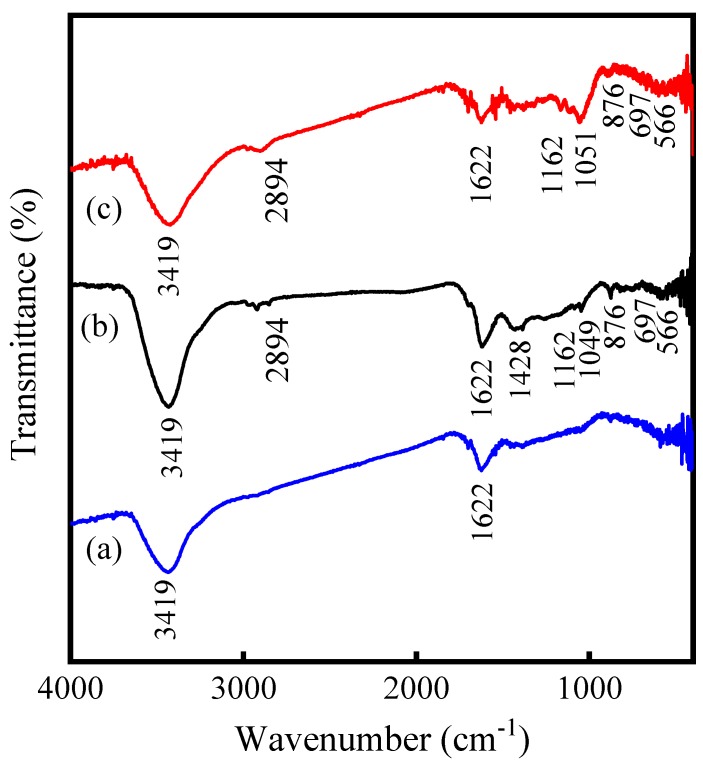
FTIR spectrum of MBC and FM-MBC. (**a**) MBC, (**b**) FM-MBC, (**c**) FM-MBC after Cr(VI) adsorption.

**Figure 9 ijerph-17-00676-f009:**
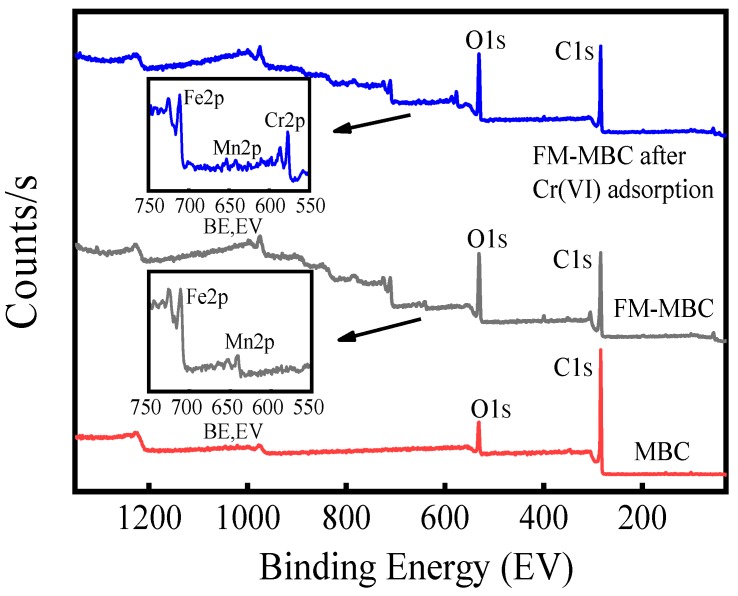
XPS of a full scan spectrum of MBC, FM-MBC and FM-MBC after Cr(VI) adsorption.

**Figure 10 ijerph-17-00676-f010:**
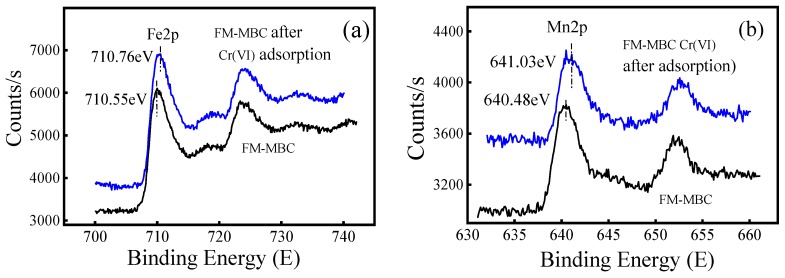
XPS spectra of FM-MBC before and after Cr(VI) adsorption, (**a**) Fe2p, (**b**) Mn2p.

**Figure 11 ijerph-17-00676-f011:**
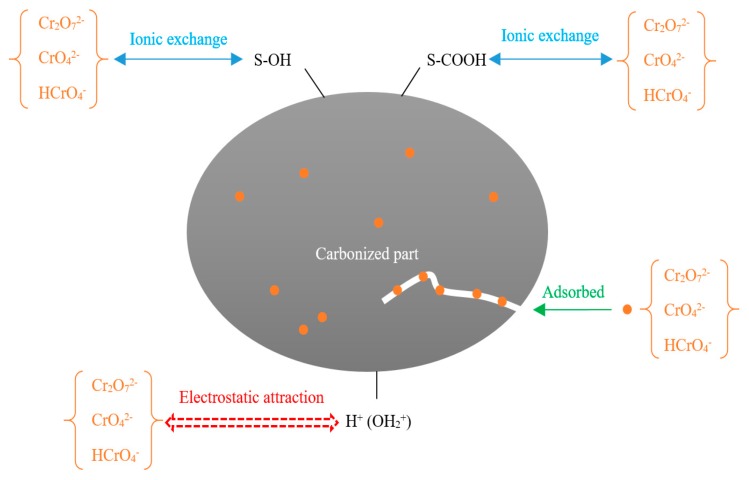
The proposed reaction mechanism of Cr(VI) on the FM-MBC.

**Table 1 ijerph-17-00676-t001:** Kinetic parameters for Cr(VI) adsorption onto the FM-MBC.

**Concentration (mg/L)**	**q_e_**	**Pseudo-First-Order Equation**			**Pseudo-Second-Order Kinetic Equation**		
	**(mg/g)**	**R^2^**	**k_1_**	**q_m_**	**R^2^**	**k_2_**	**q_m_**
20 (298 K)	9.72	0.858	0.0010	8.71	0.999	0.1002	9.68
50 (298 K)	24.39	0.786	0.0020	23.02	0.999	0.0447	24.27
50 (308 K)	24.57	0.913	0.0023	23.22	0.999	0.0420	24.43
**Concentration (mg/L)**		**Bangham Kinetic Equation**			**Elovich Kinetic Equation**		
		**R^2^**	**k_3_**		**R^2^**	**k_4_**	
20 (298 K)		0.674	0.091		0.869	0.858	
50 (298 K)		0.938	0.021		0.943	0.444	
50 (308 K)		0.923	0.032		0.925	0.723	

**Table 2 ijerph-17-00676-t002:** Isotherm parameters for Cr(VI) adsorption onto the FM-MBC.

Temperature °C	Langmuir Equation			Freundlich Equation		
	R^2^	K_L_	Q_max_ (mg/g)	R^2^	K_F_	1/n
25 (298 K)	0.999	1.107	28.49	0.875	17.274	0.357
35 (308 K)	0.999	1.215	31.21	0.866	19.038	0.350
45 (318 K)	0.999	1.224	37.62	0.871	22.301	0.341

**Table 3 ijerph-17-00676-t003:** Surface composition of the FM-MBC and FM-MBC after Cr(VI) adsorption by EDS analysis.

Element	C	O	Mn	Fe	Cr	Other
FM-MBC (wt. %)	24.33	28.10	7.62	38.01	-	1.94
FM-MBC after Cr(VI) adsorption (wt. %)	13.89	4.21	6.42	49.32	22.32	3.84

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
