# Peer review of "Preparation and Characterization of Fe-Mn Binary Oxide/Mulberry Stem Biochar Composite Adsorbent and Adsorption of Cr(VI) from Aqueous Solution"

_ijerph, 2020, doi:10.3390/ijerph17030676_

Round 1

Reviewer 1 Report

The present paper reports the adsorption property of Fe-Mn binary oxide/mulberry stem biochar composite for removing Cr(VI) species from aqueous solution. The authors insist the hybridization effect of Fe and Mn species with carbon support. The topic is appropriate for International journal of environmental research and public health. There are several points to be considered prior to publication.

The authors tried to discuss the adsorption mechanism of Cr(VI) species by the composite adsorbent based on the characterization results of XPS, FT-IR, XRD etc. However, the discussion part is poorly organized and there were less informative conclusions. That is, the authors should propose the specific adsorption site of adsorbent and reaction scheme (chemical formula). I could not understand the role of carbon support for Cr(VI); what role of hydroxyl, carboxyl groups are? The authors should estimate the kinetic constant and activation energy from the result of kinetic curve. There several careless mistakes in the main text.

Reviewer 2 Report

The paper present the preparation and characterization of Fe-Mn binary oxide/mulberry stem biochar composite adsorbent and adsorption of Cr(VI) from aqueous solution. I think the paper has a certain of innovation and carefully  research plan. There are some small problems in this paper.

In the “Abstract”, the sentence of “The aromatic groups, carboxyls, the hydroxyl groups and the hydroxyl groups were the mainly functional groups in the adsorption of Cr (VI)” has two “hydroxyl groups”. Page 2, paragraph 2, line 8: Whether is appropriate of the “……. sorption of As comparing with biochar …..”. Page 2, paragraph 3: The expression of “…the adsorption capacity of Cr(VI) by FM-MBC and MBC” may be “....the adsorption capacity of FM-MBC and MBC to Cr(VI). In section of 3.1.1, should the BET-N2 BET-N2?   Please note of Cr2O72, CrO42- and pHzpc in  2 section. In adsorption experiment, I think its adsorption capacity of FM-MBC to Cr(VI) should be tested using Cr(VI) containing wastewater based on the experimental results.

Reviewer 3 Report

Present manuscript reports the different biochar adsorbent using multiple activation techniques. The manuscript has been written well with scientific soundness and suits to the scope of the journal. Also, the contents provided have strong merits to be published. However, before accepting the article for publication, I would ask authors to respond to a few of my minor suggestions which are as follows:

Information on what was the motive, pros, and cons of authors to use and design Fe-Mn binary oxide/mulberry stem biochar composite  for Cr(VI) should be discussed?

Abstract reflects the findings reported in the manuscript. Introduction is also quite detailed , The referee wanders about many 3D carbon based adsorbent in literature, ACS Applied Materials and Interfaces, 2019, 11, 18165-18177; ACS Sustainable Chemistry & Engineering, ACS, 2019, 7, 3772-3782 ;  ACS Applied Materials and
Interfaces
, 2019, 11, 43949-43963; etc. which could have been referenced too. 

The experimental part is quite detailed which is good. 

What could be the adsorption mechanism for the Cr(VI).

Round 2

Reviewer 1 Report

I understand the authors's reply and appreciate the revision.